# Application of Response Surface Methodology to Optimize Solid-Phase Extraction of Benzoic Acid and Sorbic Acid from Food Drinks

**DOI:** 10.3390/foods11091257

**Published:** 2022-04-27

**Authors:** Bochra Bejaoui Kefi, Sana Baccouri, Rachel Torkhani, Sidrine Koumba, Patrick Martin, Naceur M’Hamdi

**Affiliations:** 1Laboratory of Useful Materials, National Institute of Research and Pysico-Chemical Analysis (INRAP), Technopark of Sidi Thabet, Ariana 2020, Tunisia; sbaccouri2@gmail.com; 2Department of Chemistry, Faculty of Sciences of Bizerte, University of Carthage, Bizerte 7021, Tunisia; 3National Institute of Research and Pysico-Chemical Analysis (INRAP), Technopark of Sidi Thabet, Ariana 2020, Tunisia; torkhanirachel@yahoo.fr; 4Transformation and Agro-Resources Unit, ULR7519, Université d’Artois-Uni LaSalle, 62408 Bethune, France; sidrine.koumbaibinga@univ-artois.fr; 5Research Laboratory of Ecosystems & Aquatic Resources, National Agronomic Institute of Tunisia, Carthage University, 43 Avenue Charles Nicolle, Tunis 1082, Tunisia; naceur_mhamdi@yahoo.fr

**Keywords:** benzoic acid, sorbic acid, food preservatives, solid-phase extraction, response surface method, Box–Behnken design, HPLC–DAD

## Abstract

An experimental design was applied for the optimization of the extraction process of two preservatives, benzoic and sorbic acids (BA, SA), from food drinks. A simple, rapid, and reliable solid-phase extraction (SPE) method for the simultaneous extraction of these two preservatives and their determination by liquid chromatography with a diode array detector was considered. Box–Behnken design (BBD) was applied to both steps of the SPE process: (i) the sample percolation to ensure the retention of the totality of the acids by the silica-based C18 sorbent; (ii) the elution step to ensure desorption of the totality of the acids from the cartridge. Thus, the volume, pH, and flow rate of the sample, and the percentage of MeOH, volume, and flow rate of the elution solvent, were optimized. Sample volume and pH have a significant influence (*p* < 0.0001 and *p* = 0.0115) on the percolation yield. However, no effect was recorded for the flow rate (*p* > 0.05). Flow rate also has no significant effect on the elution efficiency. The proposed new solid-phase extraction method, which can be easily applied to routine monitoring of preservatives BA and SA in juice and soft drink samples, included 0.5 g of C18 sorbent, 1 mL of food drink adjusted to pH 1 and percolated at 4.5 mL min^−1^, and 1 mL of a solvent mixture composed of methanol/acidified water (pH = 2.6) (90:10, *v*/*v*) used in the elution step at a flow rate of 4.5 mL min^−^^1^. Validation of the SPE method and the technique of analysis were evaluated, namely, the accuracy, precision, detection, and quantification limits and linearity. Recovery percentages of benzoic and sorbic acids were above 95% with relative standard deviations lower than 1.78%. Detection and quantification limits were 0.177 and 0.592 µg mL^−1^, and 0.502 and 0.873 µg mL^−1^ for benzoic acid and sorbic acid respectively. Optimal conditions were applied to commercial fruit juices and soft drinks and a minimal matrix effect was observed. This method was compared with other SPE methods using oxidized activated carbon and multiwalled carbon nanotubes as adsorbents. The yields determined with these last two were low compared to those determined with our method.

## 1. Introduction

Food additives (colorants, preservatives, antioxidants, stabilizers, gelling agents, thickeners, flavors, flavor enhancers, sweeteners) are added to food to facilitate their manufacture, their conservation, and modify their nutritional qualities [1,2]. However, it becomes impossible not to consume them, especially with the emergence of industrial ready-cooked food and the expanding availability of out-of-home meals and drinks [3,4,5]. Preservatives are food additives added to a wide range of commercial food products and have a considerable role in the food industry [6,7]. On the other hand, they must be used at certain levels because they can present a danger to the health of consumers [6,7,8]. Benzoic and sorbic acids (BA, SA) are used as food additives and preservatives to prevent food spoilage and extend shelf life [9,10]. They proved efficient in the preservation of food products especially since they are stable during food preparation and processing [11,12,13,14]. The maximum permitted level in soft drinks at 150 mg L^−1^ is defined by the FAO/WHO Expert Committee. The characterization of these preservatives is important, whether in foods [15], cosmetics [16,17], or pharmaceuticals [18,19], not only for the quality control of the product but also for the number of allergies for which preservatives are responsible [20,21]. As a result, regulators’ institutions defined dose limits and even impose control of these compounds to protect consumers. Therefore, the food industry requires the development and validation of rapid and accurate analytical methods to verify the levels of added preservatives.

To assess their levels in foodstuffs, it is important and necessary to develop effective, reliable, practical, inexpensive, and rapid analytical methods. In many cases, before analysis, sample preparation is essential to isolate and preconcentrate the analytes from the complex food matrix to increase the sensitivity of the analysis method. However, sample preparation is often the most critical part of the preservative analysis, which relies mainly on the physicochemical properties of the analytes and the composition and complexity of food samples. More than 60% of the analysis time is attributable to sample preparation and about 30% of the experimental errors are inherent to the sample processing [22,23]. In this context, several sample processing methods for the extraction and preconcentration of preservatives are applied, namely, the complexation and chemical precipitation reactions, membrane filtration, liquid–liquid extraction, and solid-phase extraction (SPE) [24,25,26,27,28]. Among cited methods, SPE can be considered as one of the fastest and simplest sample preparation techniques for extracting trace-level analytes from liquid samples. It has been reported to be an efficient, ecofriendly, rapid, and economical technique [29,30,31,32,33]. When this method is optimized with experimental design methodology, it gives precise, accurate, and reproducible results. Statistical experimental designs provide more reliable results and a robust sample processing method. This methodology is increasingly replacing the classical optimization (one factor at a time), which can in some cases lead to erroneous results if there are interactions between the different experimental factors. The experimental design methodology assesses the effects of multilevel factors, together with the interaction between them [15,31].

The aim of this study was to develop and optimize a new and efficient solid-phase extraction method that could be easily adopted for routine monitoring of benzoic and sorbic acids preservatives in juice and soft drink samples. The potential factors affecting the SPE of the analytes were optimized using the experimental design methodology to obtain the best extraction performance and maximum extraction yields. The sorption and desorption steps were evaluated separately and optimized using Box–Behnken designs. A comparison between the C18 sorbent used and the activated carbon and multiwalled carbon nanotube adsorbents, for the extraction of the two acids from drink samples before their simultaneous analysis by liquid chromatography, was investigated. Furthermore, the efficiency of the optimized SPE method for the enrichment of selected preservatives in fruit juice samples and soft drinks was studied.

## 2. Materials and Methods

### 2.1. Reagents and Solutions Preparation

All reagents were of analytical grade. Methanol LC grade and glacial acetic acid were purchased from Merck (Darmstadt, Germany) and Fisher Chemical, respectively. Ammonium acetate (Loba Chemie PVT L.T.D, 96% purity, 0.01 mol L^−1^), concentrated hydrochloric acid (Panreac Quimica, 37%), and sodium hydroxide (Fisher Chemical, purity ≥ 99%, 1 mol L^−1^) were used. Certified standard solutions of BA and SA were obtained from Fluka (purity ≥ 99%). Standard stock solutions of 1 g L^−1^ of both acids were prepared in methanol/ultrapure water (40:60, *v*/*v*). Standard mixtures, ranging from 1 to 45 µg mL^−1^, were prepared by appropriate dilution of the stock solution with acetate buffer/methanol (60:40, *v*/*v*) [34]. The buffer solution (pH = 4.6) was prepared by mixing ammonium acetate and acetic acid in appropriate proportions. Silica-based sorbent with octadecyl functional group was acquired from Applied Separations Company, Allentown, PA, USA (3 mL; 500 mg, with a particle size of 40 μm and an average porosity of 60 A°), and multiwalled carbon nanotubes (MWCNT) were purchased from Sigma-Aldrich (France). MWCNT were 5–9 µm in length and 110–170 nm external diameter. Activated carbon (AC) was purchased from Sigma (20–60 mesh, product number C3014).

### 2.2. Apparatus

The chromatographic analyses of BA and SA were carried out using a 1100 LC system (Agilent Technologies) equipped with a quaternary pump (model G1311A), degasser (model G7122A), automatic injector (model G1313A), diode array detector DAD (model G1315B), and Agilent ChemStation data processing software (Rev. B.02.01). Chromatographic separation was performed on an Agilent Zorbax SB-C8 analytical column (150 mm × 4.6 mm i.d., 5 µm particulate sizes), which were maintained at a temperature of 25 °C. The mobile phase consisted of 55.55% acetate buffer and 44.44% methanol adjusted to pH 4.5 with acetic acid [35]. The separation was performed in isocratic mode, the flow rate was 0.7 mL min^−1^, and the injection volume was 10 µL. The identification was carried out using a DAD at a wavelength of 235 nm.

### 2.3. Solid-Phase Extraction Procedure

The solid-phase extraction of benzoic and sorbic acids was performed using a vacuum extraction module. Before loading samples, the cartridge was first activated with 10 mL of methanol and conditioned with 10 mL of UP water. After that, a volume of UP water spiked with BA and SA at 40 µg mL^−1^ was percolated through the cartridges. The volume, pH, and flow rate of the percolated sample were optimized to ensure the maximum retention of the two acids on the solid phase. The pH of samples was set between 1 and 3 with HCl or NaOH solutions. The solutes were then eluted with methanol/acidified water (1% acetic acid, pH = 2.6). The elution volume, percentage of MeOH, and the elution flow rate was also optimized to ensure desorption of all solutes from the SPE cartridge. These two SPE steps were optimized using the Box–Behnken design.

### 2.4. Experimental Design

Response surface methodology (RSM) was the technique applied for the optimization of the analytical procedure, and a Box–Behnken design was the model used [15,31,34,35,36,37]. Three SPE factors were defined for the percolation and elution steps, and the number of experiments required was calculated by applying the following BBD equation:(1)$$N=2K\left(k-1\right)+C_{p}$$
where *N* is the total number of experiments required; *k* is the number of factors (3 factors: X_1_, X_2_, X_3_); and *Cp* is the center point.

Accordingly, a design consisting of 15 experimental points including three center points was used to assess the effects of three variables and the interaction effects on responses by fitting the data to a polynomial model. The three most important parameters of each step (percolation and elution) were chosen as independent variables and named X_1_, X_2_, and X_3_, and set at three coded levels (−1, 0, +1). Design-Expert 13.0 software was used in the experimental design model building and data analysis.

### 2.5. Application of BBD for the Optimization of Percolation and Elution Steps

The two steps of the SPE method, sample percolation and acids elution, were optimized using the BBD. Specific volumes (factor X_1_) of the standard solution (40 µg mL^−1^) were prepared and adjusted to the desired pH (factor X_2_) and then percolated through the SPE cartridge at a flow rate (factor X_3_) chosen by the percolation BBD. After passing the acids solution through the column, the recovered solution (reject) was analyzed by HPLC–DAD to determine the amount of acids adsorbed onto silica-based C18. Optimal conditions of this step were then applied when the next elution step was optimized. Indeed, the acids retained on the cartridge were eluted with methanol/acidified water at proportions (factor X1), volumes (factor X_2_), and flow rate (factor X_3_) experimentally chosen according to the BBD. The eluted solution (extract) was then analyzed by HPLC–DAD to determine concentrations of the eluted acids. These eluted acids give the extraction yield of the SPE method. Studied responses are the retention yields (*Rr*) and elution yields (*Re*) of benzoic and sorbic acids. These yields are calculated as follows:(2)$$R_{r}=\frac{Q_{r}}{Q_{0}}=\frac{Q_{0}-Q_{exp}}{Q_{0}}$$
(3)$$R_{e}=\frac{Q_{e}}{Q_{0}}$$
where $Q_{0}$ is the initial concentration of the acid in the percolated solution (40 µg mL^−1^); $Q_{r}$ is the concentration of the acid retained on the adsorbent during the percolation step; $Q_{exp}$ is the concentration of the acid determined in the recovered solution obtained after the percolation step; and $\;Q_{e}$ is the concentration of the acid determined in the extract after the elution step.

These responses were predicted using the generalized second-order model given in the following equation:(4)$$R=\beta_{0}+\sum_{i=1}^{3}\beta_{i}x_{i}+\sum_{i=1}^{3}\beta_{ii}x_{i}{}^{2}+\sum_{i=1}^{2}\sum_{j=i+1}^{3}\beta_{ij}x_{i}x_{j}+\varepsilon$$
where ***R*** is the predicted response, $\beta_{0}$ is a constant, $\;\beta_{i}$ is the linear effect of variable $X_{i}$, $\;\beta_{ii}$ is the quadratic effect of $X_{i}$, $\;\beta_{ij}$ is the linear interaction effect of $X_{ij}$, and ε is a statistical error term.

Table 1 contains coded values and factor levels of the two BBDs applied to percolation and elution steps.

The detailed matrix of BBD (for both percolation and elution steps) for the three-factor three-level design with the center is shown in Table 2.

### 2.6. Sample Preparation

The optimal conditions obtained with the optimized SPE method were then applied to commercial food drinks: fruit juices and soft drinks. Soft drinks were firstly degassed for 30 min and then filtered using 0.45 µm pore size membrane filters. While the samples of fruit juices are well homogenized to reduce to the maximum the pulp, if present. Then, 5 mL of each sample was diluted to 100 mL in ultrapure water and the pH was adjusted to 1 with hydrochloric acid. Finally, the SPE method was applied under its optimal conditions and the extract obtained was analyzed by HPLC–DAD. The same protocol described above for unspiked samples was applied to spiked samples with 1 mL of a standard solution containing BA and SA at concentrations of 10 µg mL^−1^.

### 2.7. Comparative Study

The present SPE method was compared to other SPE protocols using oxidized activated carbon (AC) and multiwalled carbon nanotubes (MWCNT). The AC was oxidized with concentrated nitric acid HNO_3_ at 25 °C by stirring for 24 h [36]. These two adsorbents were first oven-dried at 80 °C for 2 h and then packed in empty SPE cartridges (0.2 g, 3 mL, polypropylene) and to hold their packing sorbent in place; polypropylene upper and lower frits (20 µm porosity) were placed at each end of the cartridge. Carbon nanotubes were purified with an HCl solution (2 M) and washed with UP water. Finally, the cartridges are dried to remove any remaining water or solvents. Oxidized AC and MWCNT were used as SPE adsorbents using the following experimental conditions: conditioning with 10 mL MeOH and 10 mL ultrapure water; percolation of 1 mL of a standard solution of BA and SA (10 µg mL^−1^) adjusted to pH 1; and elution with MeOH/acidified water (90:10 *v*/*v*).

## 3. Results and Discussion

### 3.1. Effect of Experimental Parameters on Retention Yields

Parameters considered for the optimization of the percolation step are the sample volume (X_1_), sample pH (X_2_), and flow rate of the percolation (X_3_). The experimental design and results are reported in Table 3. The experimental response studied is the retention efficiency, expressed in terms of percentage of retention (*Rr%*).

### 3.2. Model-Fitting and Statistical Analysis

The polynomial mathematical model developed for optimization is a second-degree model. Figure 1 and Figure 2 showing the curves of predicted versus observed values confirm the goodness of fit (0.92 and 0.98). For each coefficient in the regression model, significance was assessed by the corresponding *p* values (*p* < 0.0026) [38]. The coefficients of determination (R^2^) of the quadratic regression models vary between 0.98 (Figure 1) and 0.99 (Figure 2), and values of predicted coefficients of determination vary from 0.92 to 0.96 for SA and BA, respectively, showing a reasonable agreement with the experimental results [39].

Values of R^2^ higher than 90% indicate a good fit between experimental values with those obtained by the models. The results obtained confirm that the actual values are very close to the predicted values (0.96 and 0.99). Mean values of 6.3 and 3.8 (*p* > 0.05) were obtained for the lack-of-fit test, suggesting that the model is reliable in predicting the response and that it can predict variations within the response [37]. Figure 1 and Figure 2 confirm that the curve of observed values as a function of predicted values perfectly fits the shape of a straight line; we note the close agreement that exists between the experimental results and the theoretical values predicted by the polynomial model [40,41,42,43,44].

### 3.3. Analysis of Significant Factors

The statistical significance of each term (linear, interaction, and quadratic) has been reported in Table 4 obtained from the analysis of variance. Each of the three terms can be considered for the second-order fit. The *F*-value of models of the first-order, two-way interactions, and pure quadratic, indicate that the models were significant at *p* < 0.05 for the percolation response of both acids. The lack-of-fit values, 3.8 and 6.3 associated with *p*-values of 0.21 and 0.14, were not significant due to the relative pure error. The results of the present study show that the model is significant with *p* < 0.05, and by this, the validity of the model is confirmed. Therefore, this model could work for percolation optimization. A summary of significant factors (*p* < 0.05) and their effect on the response variable are shown in Table 4. The results obtained show that the volume and pH have a significant influence (*p* < 0.0001 and *p* = 0.0115) on the retention yield. However, no effect was recorded for the flow rate (*p* > 0.05). The sample volume (X_1_) played a primary role in improving percolation efficiency. Our results agree with other studies [38,42,45]. Overall, the statistical analysis suggests that the experimental values fit the models well, with good accuracy and reliability.

### 3.4. Mathematical Models

The coded values were analyzed using the Design-Expert software, with no transformation and quadratic model. The models showed adequate *p*-value and an insignificant lack of fit. The results obtained showed that the two models ($R_{r}{}_{BA}$ and $R_{r}{}_{SA}$) were statistically meaningful (their *p*-value < 0.000). Moreover, the coefficients R^2^ for the responses $R_{r}{}_{BA}$ and $R_{r}{}_{SA}$ were, respectively, 0.99 and 0.98. These data reveal that the correlation was good, indicating a good fit of the models. The regression equations for the response variables in terms of coded factors are:(5)$$\begin{array}{ll} R_{r}{}_{BA}= & 10.05-42.73{\;X}_{1}-4.20{\;X}_{2}+1.05{\;X}_{3}+3.06{\;X}_{1}X_{2}+2.39{\;X}_{1}X_{3}-0.19{\;X}_{2}X_{3}+38.70{\;X}_{1}{}^{2}+2.2{\;X}_{2}{}^{2}\\ & +0.1481{\;X}_{3}{}^{2} \end{array}$$
(6)$$\begin{array}{ll} R_{r}{}_{AS}= & 8.85-42.83{\;X}_{1}-3.88{\;X}_{2}+1.37{\;X}_{3}+4.26{\;X}_{1}X_{2}+3.09{\;X}_{1}X_{3}+0.046{\;X}_{2}X_{3}+39.41{\;X}_{1}{}^{2}+1.20{\;X}_{2}{}^{2}\\ & -0.0035{\;X}_{3}{}^{2}\; \end{array}$$

### 3.5. Effect of Interaction between Factors

The quadratic model obtained was used to calculate the response surface for each variable separately. Figure 3, Figure 4 and Figure 5 show the response surface of Rr as a function of each pair of the independent variables. In Figure 3, the response model is mapped against sample volume (X_1_) and pH (X_2_), while the flow rate (X_3_) is held constant at its central level. Examination of the results shows that retention yields increase when pH and sample volume decrease. As shown in Figure 3, the retention yields reached a maximum value at pH 1 and a sample volume of 1 mL.

Figure 4 shows the effect of sample volume (X_1_) and flow rate (X_3_) on the retention yields; pH is kept constant at 1. The diagnosis of response surfaces shows that the mutual interaction between the percolation volume and the flow rate is not significant (*p* > 0.001). The response surface plots show a significant effect of sample volume on percolation efficiency. However, no significant effect was observed for the sample flow rate (*p* > 0.001).

From response surface plots illustrated in Figure 5, we observe that the interaction between pH and the flow rate is not significant (*p* > 0.001).

### 3.6. Effect of Experimental Parameters on Elution Yield

After optimizing the percolation step, a second BBD with three factors and three center points was used to optimize the elution step. Independent variables were the percentage of MeOH % in the elution solvent (X_1_), the elution volume (X_2_), and the elution flow rate (X_3_). Values of the elution yields of BA and SA did not follow the normal distribution. A logarithmic transformation of the results (log Re) was, therefore, necessary to apply the BBD to calculated elution yields. The experimental design and elution yields are shown in Table 5.

### 3.7. Model Adjustment

The goodness of fit of the models was evaluated by the adjusted determination and the predicted determination coefficients. Polynomial mathematical models developed for optimization are second-degree models. The plot of predicted versus observed values (Figure 6 and Figure 7) confirms the good fitting ability. Adjusted coefficients of determination of the models vary between 0.92 and 0.98, and values of predicted coefficients of determination vary from 0.82 to 0.92. Results obtained confirm that real values are very close to predicted values. Therefore, these models can be used to optimize responses due to the high correlation between observed and predicted values [37].

### 3.8. Analysis of Significant Factors

Table 6 reports the ANOVA results. The analysis of variance shows that the overall models are significant with *p* < 0.05, confirming the validity of these models for elution optimization. The results obtained show that the volume and % MeOH have significant effects (*p* < 0.05) on retention yield. On the other hand, no effect was recorded for the flow rate (*p* > 0.05). The *p*-values of linear coefficients (X_1_ and X_2_), interaction coefficients (X_1_X_2_), and quadratic coefficients (X_2_^2^) were less than 0.05, which indicates significant effects on elution efficiency. Coefficients in the equations ($R_{e}{}_{\mathrm{BA}}$, $R_{e}{}_{\mathrm{SA}})$ provide insight concerning the effects and the interaction between the factors that were studied (% MeOH, elution volume, and flow rate). The most important variables are the volume of the solvent, the percentage of methanol in the solvent, and their interactions. However, the elution flow rate has a negligible effect on the elution yield for both acids.

### 3.9. Mathematical Models

From the validation of the parameters studied (X_1_, X_2_, and X_3_), the second-order polynomial was obtained, and it describes the response surface. The final equations, using the retention yield as a response for BA and SA, are, respectively:(7)$$\begin{array}{ll} R_{e}{}_{\mathrm{BA}}= & 0.21+0.09X_{1}-0.94X_{2}+0.004X_{3}-0.155X_{1}X_{2}-0.024{\;X}_{1}X_{3}+0.036X_{2}X_{3}-0.046{\;X}_{1}{}^{2}+0.38{\;X}_{2}{}^{2}\\ & +0.07{\;X}_{3}{\;}^{2} \end{array}$$
(8)$$\begin{array}{ll} R_{e}{}_{\mathrm{SA}}= & 0.55+0.082{\;X}_{1}-0.91{\;X}_{2}-0.0017{\;X}_{3}-0.127X_{1}X_{2}+0.003{\;X}_{1}X_{3}+0.056{\;X}_{2}X_{3}-0.025{\;X}_{1}{}^{2}\\ & +0.37X_{2}{\;}^{2}+0.054{\;X}_{3}{\;}^{2} \end{array}$$

The validity of the model was determined by ANOVA. The coefficients of determination obtained were 0.92 and 0.98 for BA and SA, respectively, which indicates good agreement between the experimental and predicted values. The value of the F-test indicates that the second-order model was statistically significant (62.9 > 9.01).

### 3.10. Interaction Effects

The 3D response surface plots showing the elution yields against individual factors are shown in Figure 8, Figure 9 and Figure 10, which illustrate the interaction of the volume of eluent with the percentage of MeOH, eluent flow rate with its volume, and the interaction of eluent flow rate with the percentage of MeOH, respectively. Results show an enhanced analytical response (Re %) when the percentage of MeOH is between 50 and 90 and the volume of eluent values is between 1 and 10 mL. Response surface plots reveal that the analytical response increased with decreasing volume of the eluent and increasing MeOH percentage. Maximum elution yields were determined at eluent volumes and MeOH percentage close to 1 mL and 90%, respectively. Furthermore, the elution is enhanced when the elution flow rate is in the middle of its experimental values.

### 3.11. Determination of Optimal Conditions

Different factors can affect the SPE efficiency; therefore, their optimization through a multivariate approach is recommended, especially when these factors are correlated [46]. According to the results of the optimization of percolation and elution steps and desirability studies, experimental conditions given in Table 7 were chosen as optimal values for the SPE extraction of both studied acids. Similar results have been found in previous studies [41,47].

### 3.12. Method Validation

To consider all the concentrations of analyzed acids, during the optimization of the SPE method, several calibration ranges were considered (1–5, 1–15, and 25–45 µg mL^−1^). Three calibration curves were thus drawn for the two acids. Equations of the curves and the coefficients of determination R^2^, are given in Table 8. The R^2^ values are greater than 0.9; we can therefore conclude that the regression models applied are linear. The LDD and LDQ were determined from regression lines determined for AB and AS (calibration range 1–5 µg mL^−1^, repeated 3 times) and Y-intercepts were considered as blank responses. LDD and LDQ were calculated as shown below, and values are given in Table 8 [48].
(9)$$\mathrm{LDD}/\mathrm{LDQ}=\frac{k\;s}{a}$$
where: k—factor of 3 and 10 for LDD and LDQ, respectively; s—standard deviation of the Y-intercept; and a—slope of the regression line.

The precision of this method was studied [48]. For this purpose, five extractions of the ultrapure water spiked with the two acids at 5 µg mL^−1^ were performed using the optimum conditions described above. The repeatability of the SPE method was determined through the CV of the mean concentration obtained from the analysis of the five replicates achieved over a single day, while the reproducibility was calculated with results performed over different days and the CV of the mean concentration was calculated. The mean recovery percentages (~95%) showed a CV lower than 1.78% for the two acids, which highlights the high accuracy of the method. The data were also characterized by their relatively high precision. Indeed, CV values for intraday precision for AB and AS were lower than 1.48% and 2.34%, respectively. However, CV values for interday precisions for AB and AS were 1.24% and 1.85%, respectively.

### 3.13. Application to Real Samples

Robustness is the ability of an analytical method to provide small variations in results when subjected to controlled changes in application conditions (NF V 01—000). To verify the effect of a complex matrix and the robustness of our SPE method, all retained SPE conditions were assessed on real samples (three fruit juices and three soft drinks). Doped (with the two acids at 10 µg mL^−1^) and nondoped samples undergo the optimized SPE extraction method. The extracts obtained were analyzed by HPLC–DAD. The relative chromatograms are shown in Figure 11 and Figure 12. As for extraction, yields are calculated as follows and shown in Table 9:(10)$$R\%=\frac{C_{2}-C_{1}}{C_{0}}*100$$
where C_2_ is the concentration of the acid in the extract obtained from a spiked sample, C_1_ is the concentration of the acid in the extract obtained from a nonspiked sample, and C_0_ is the concentration of the acid added to the sample (spiking level: 10 µg mL^−1^).

The recoveries of benzoic and sorbic acids from real samples (Rs) are between 80 and 99.51% (Table 9). These yields correspond well to the model prediction and are very close to those determined during the SPE method optimization. Therefore, an insignificant effect of the matrix on the efficacy of our extraction method was observed. These results also show the importance of the application of an experimental design for the optimization of an experimental method or process.

### 3.14. Comparison Study

The present optimized SPE method was compared with an SPE method using an abundant and inexpensive adsorbent, oxidized activated carbon (AC), and with an SPE using multiwalled carbon nanotubes, a synthetic and expensive nanomaterial [47,48]. Table 10 shows the average recovery percentages with the relative standard deviations of the three SPE methods tested.

Yields determined with both AC and carbon nanotubes adsorbents are low compared to those determined with silica-based C18. The oxidation of coal generally increases the carboxyl, lactone, phenolic, and carbonyl groups on the surface of this material. Carboxyl groups tend to form water clusters on the micropore openings in the AC surface, thus blocking benzoic and sorbic acids from entering the micropores and hindering the process of adsorption [39]. As for multiwalled carbon nanotubes, their morphology is nanotubular and multiwalled. This morphology is neither homogeneous nor regular, which does not ensure good reproducibility of extraction yields (RSD > 5%).

## 4. Conclusions

In the present study, a novel SPE method was developed for the enrichment of benzoic and sorbic acids from food drinks, to be analyzed by liquid chromatography. Therefore, this method could be easily adopted for routine monitoring of BA and SA preservatives in juice and soft drink samples. Compared to the usual method (International Standard ISO 22855-2008) [33], the proposed procedure is simple, fast, clean, and reliable. The extraction and preconcentration of these two acids before analysis increased the sensitivity of the detection method by reducing the matrix effect and concentrating these two acids in the final extract. On the other hand, the optimization of this SPE method by experimental design methodology (BBD) allowed for obtaining a robust, reliable, reproducible, and precise method with maximum extraction yield. The results showed good extraction yields (higher than 95%) under the following conditions: Conditioning of the C18 column was performed with 10 mL methanol followed by 10 mL of UP water. Then, 1 mL of sample, adjusted to pH 1, was percolated at a flow rate of 4.5 mL min^−1^. Finally, the elution of acids was done with 1 mL of methanol/acidified water (pH = 2.6) (90:10, *v*/*v*) at a flow rate of 4.5 mL min^−1^. Optimal conditions thus determined were successfully applied to commercial fruit juices and soft drinks. A slight matrix effect was observed since the calculated yields were close to those determined during the optimization. The present method was compared with SPE using an oxidized carbon AC and multiwalled carbon nanotubes MWCNT. Silica-based C18 presented better extraction yields. Thus, the proposed new solid-phase extraction method could be used for routine monitoring of preservatives BA and SA in juice and soft drink samples.

## Figures and Tables

**Figure 1 foods-11-01257-f001:**
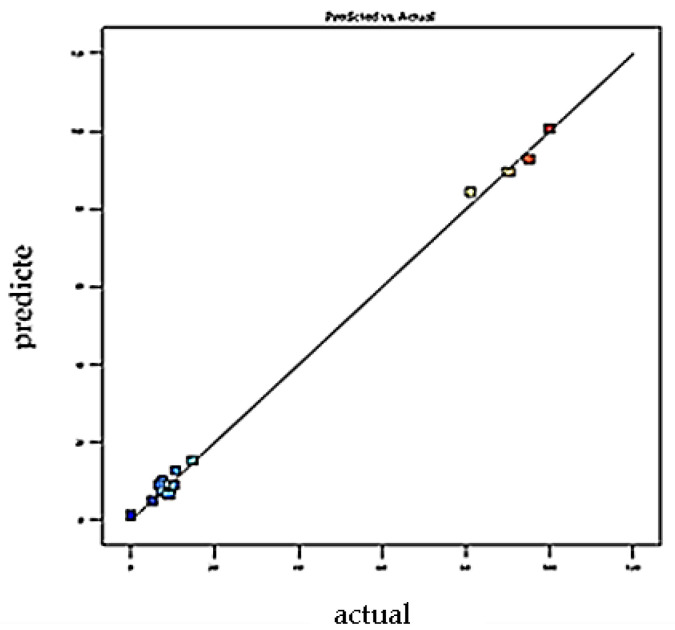
Plot of predicted value vs. actual value for SA retention efficiencies. Y _Predicted_; *p* < 0.0026; R^2^ _adjusted_ = 0.98; R^2^ _predicted_ = 0.92.

**Figure 2 foods-11-01257-f002:**
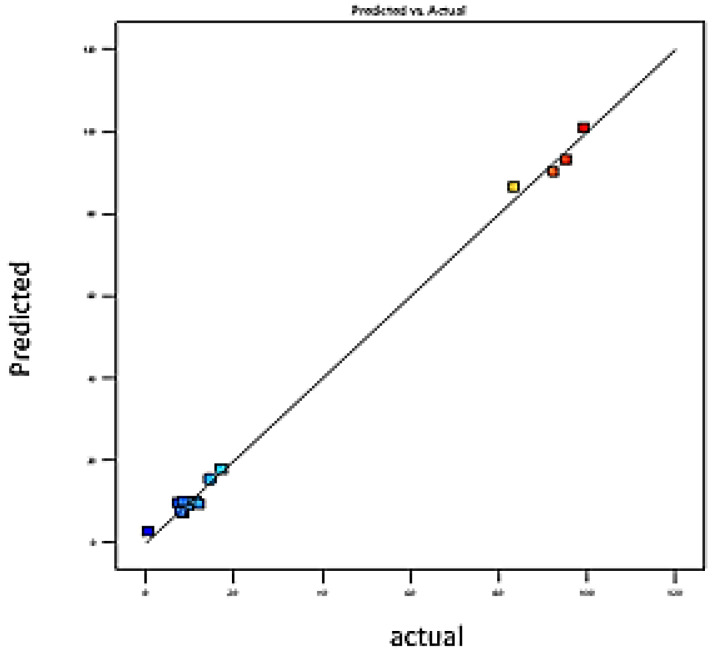
Plot of predicted value vs. observed value for BA retention efficiencies. Y_Predicted_; *p* < 0.0001; R^2^ _adjusted_ =0.99; R^2^ _predicted_ = 0.9.

**Figure 3 foods-11-01257-f003:**
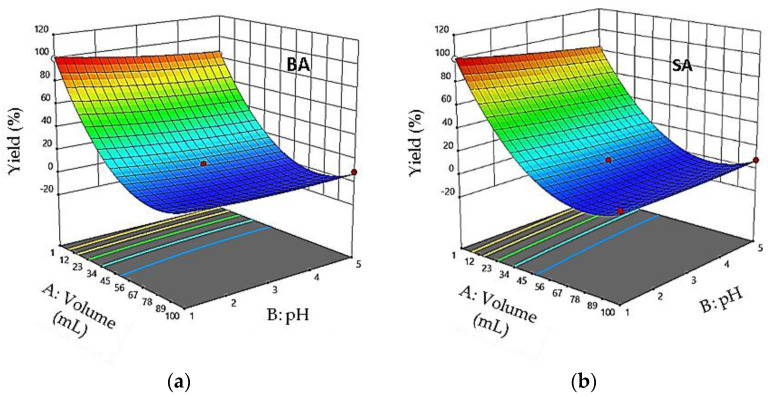
Response surface plots showing the effect of the sample volume and pH on percolation efficiency: flow rate = 4.5 mL min^−1^. (**a**) $R_{r}{}_{BA}$; (**b**) $R_{r}{}_{SA}$.

**Figure 4 foods-11-01257-f004:**
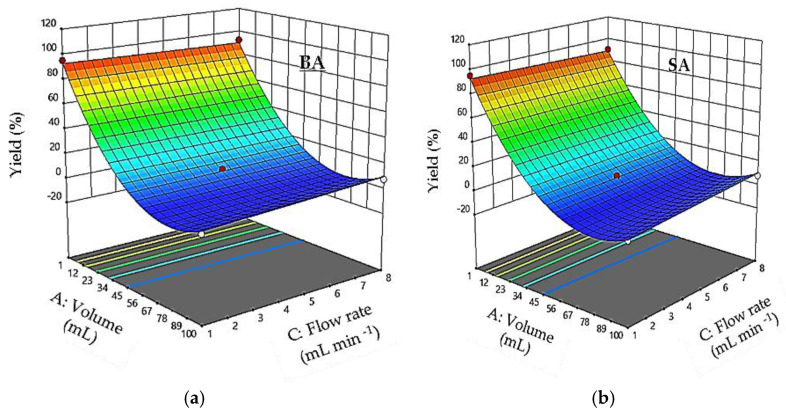
Response surface plots showing the effect of the sample volume and flow rate on percolation efficiency: pH = 1. (**a**) $R_{r}{}_{BA}$ and (**b**) $R_{r}{}_{SA}$.

**Figure 5 foods-11-01257-f005:**
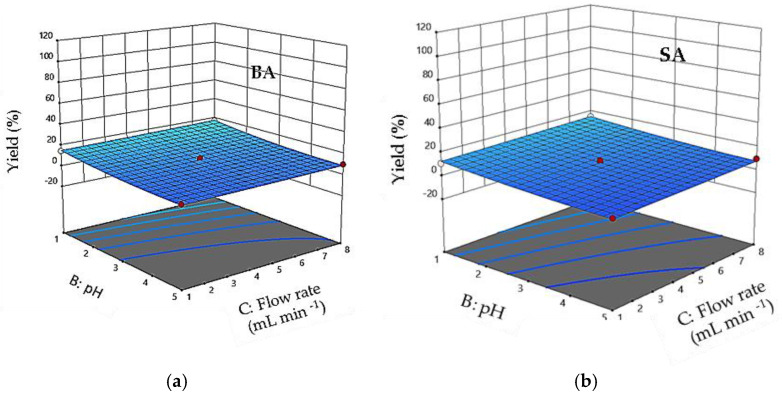
Response surface plots showing the effect of the sample pH and flow rate on percolation efficiency: sample volume = 1 mL. (**a**) $R_{r}{}_{BA}$; (**b**) $R_{r}{}_{SA}$.

**Figure 6 foods-11-01257-f006:**
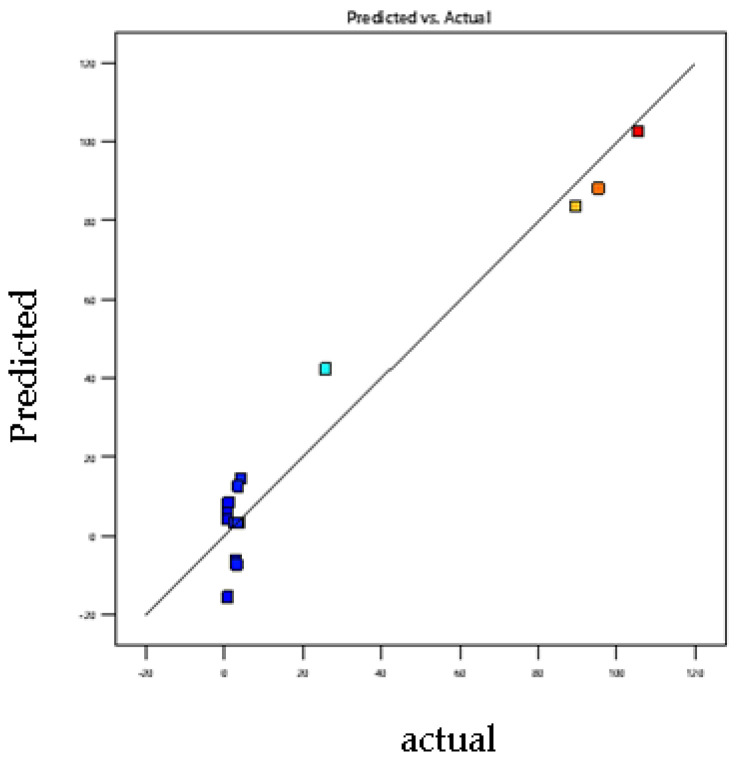
Plot of predicted value vs. observed value for BA elution efficiencies. Y _Predicted_
*p* < 0.023; R^2^
_adjusted_ = 0.98; R^2^
_predicted_ = 0.92.

**Figure 7 foods-11-01257-f007:**
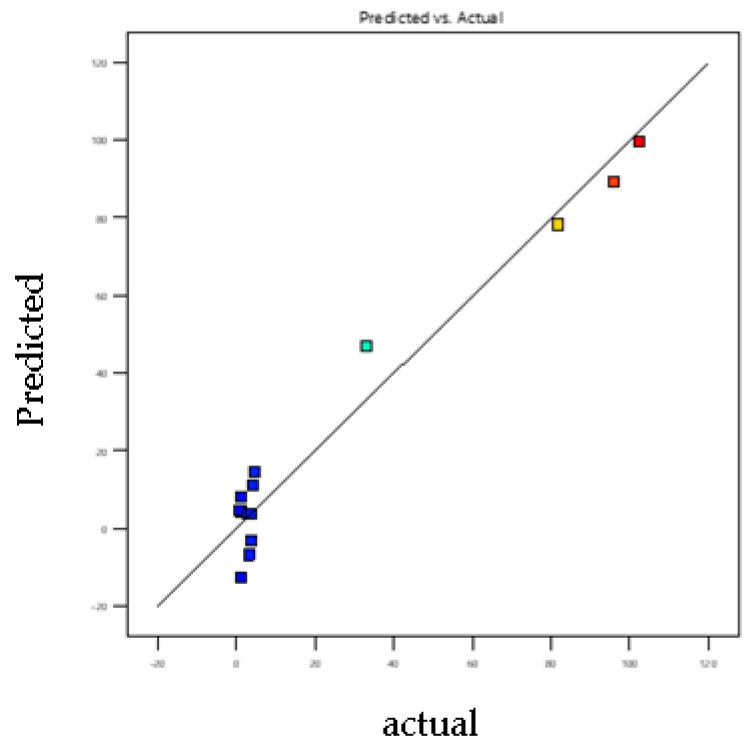
Plot of predicted value vs. observed value for SA elution efficiencies. Y _Predicted_
*p* < 0.04 R2 _adjusted_ = 0.92, R^2^
_predicted_ = 0.82.

**Figure 8 foods-11-01257-f008:**
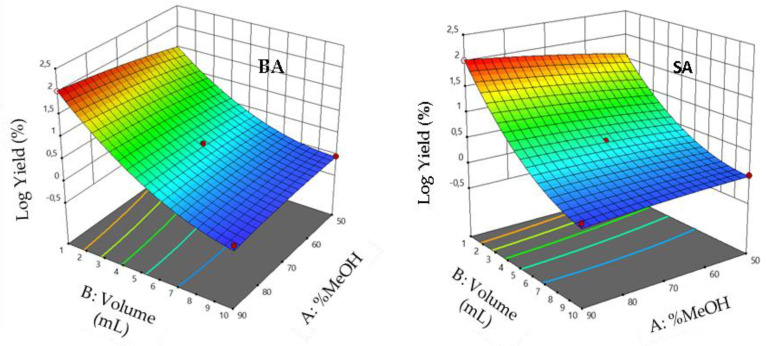
Response surface plots showing the effect of the MeOH percentage and solvent volume on elution efficiency. The flow rate is fixed at the optimum (**left**) $R_{r}{}_{BA}$ and (**right**) $R_{r}{}_{SA}$.

**Figure 9 foods-11-01257-f009:**
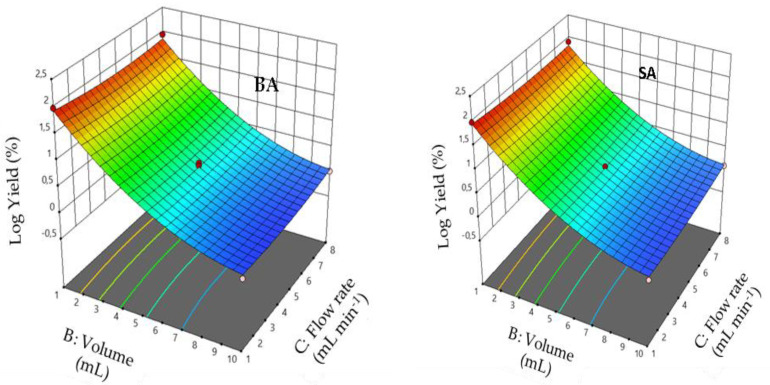
Response surface plots showing the effect of solvent volume and flow rate on elution efficiency. MeOH percentage is fixed at the optimum (**left**) $R_{r}{}_{BA}$ and (**right**) $R_{r}{}_{SA}$.

**Figure 10 foods-11-01257-f010:**
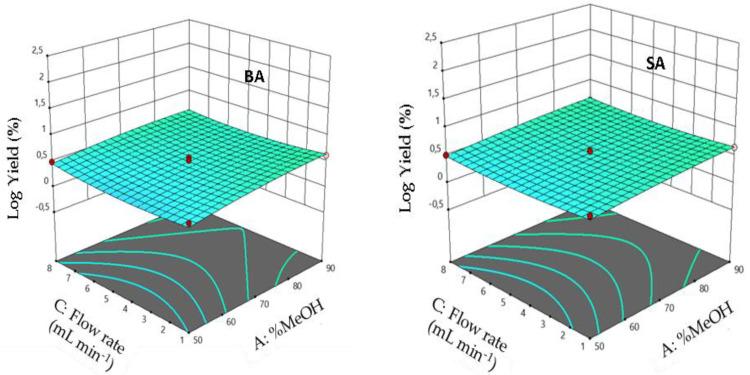
Response surface plots showing the effect of the MeOH percentage and flow rate on elution yield. Solvent volume is fixed at the optimum (**left**) $R_{r}{}_{BA}$ and (**right**) $R_{r}{}_{SA}$.

**Figure 11 foods-11-01257-f011:**
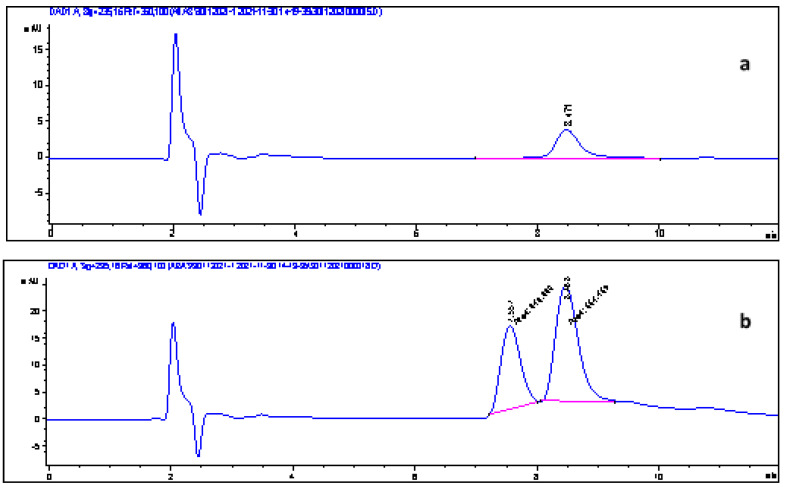
Chromatograms of the juice sample (**a**) and the juice sample spiked with 5 µg mL^−1^ of BA and SA (**b**). Benzoic acid: 7.5 min; sorbic acid: 8.4 min.

**Figure 12 foods-11-01257-f012:**
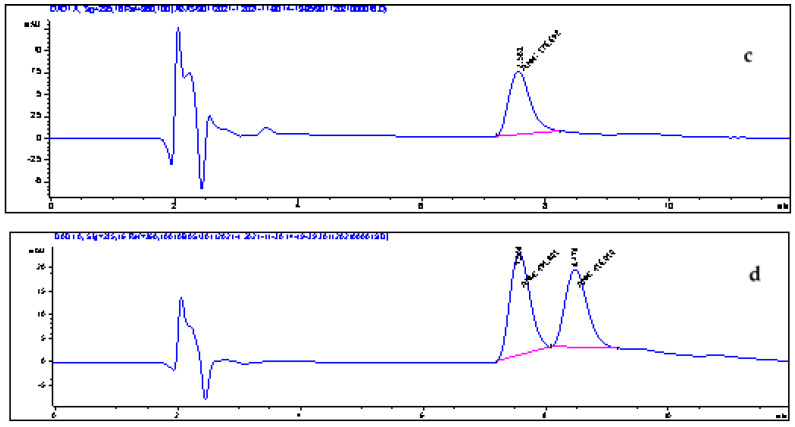
Chromatograms of the soft drink sample (**c**) and the soft drink sample spiked with 5 µg mL^−1^ of BA and SA (**d**). Benzoic acid: 7.5 min; sorbic acid: 8.4 min.

**Table 1 foods-11-01257-t001:** Investigated variables and their levels studied in the BBD of percolation and elution steps.

Factor		Level		
		Percolation		
		−1	0	+1
**X_1_**	**Sample volume (mL)**	1	50.5	100
**X_2_**	**pH**	1	3	5
**X_3_**	**Flow rate (mL min^−1^)**	1	4.5	8
		**Elution**		
**X_1_**	**Percentage of MeOH (%)**	50	70	90
**X_2_**	**Solvent elution volume (mL)**	1	5.5	10
**X_3_**	**Flow rate (mL min^−1^)**	1	4.5	8

**Table 2 foods-11-01257-t002:** The three-factor three-level BBD design.

Trial Run	X_1_	X_2_	X_3_
1	−1	−1	0
2	+1	−1	0
3	−1	+1	0
4	+1	+1	0
5	−1	0	−1
6	+1	0	−1
7	−1	0	+1
8	+1	0	+1
9	0	−1	−1
10	0	+1	−1
11	0	−1	+1
12	0	+1	+1
13	0	0	0
14	0	0	0
15	0	0	0

**Table 3 foods-11-01257-t003:** Experimental plan and results of the BBD for the SPE percolation step.

Experiment	Experimental Plan			BA	Rr (%)
	X_1_ (mL)	X_2_	X_3_ (mL min^−1^)		SA
1	1	1	4.5	99.41 ± 5.57	99.99 ± 5.68
2	100	1	4.5	12.24 ± 0.23	9.05 ± 0.42
3	1	5	4.5	83.54 ± 3.24	81.34 ± 4.73
4	100	5	4.5	8.59 ± 0.12	7.46 ± 0.11
5	1	3	1	95.20 ± 5.01	95.10 ± 5.13
6	100	3	1	0.56 ± 0.01	0.00 ± 0.00
7	1	3	8	92.45 ± 4.72	90.33 ± 4.35
8	100	3	8	7.38 ± 0.11	7.59 ± 0.13
9	50.5	1	1	14.64 ± 0.18	10.75 ± 0.62
10	50.5	5	1	7.97 ± 0.10	5.26 ± 0.46
11	50.5	1	8	17.20 ± 0.25	14.74 ± 0.22
12	50.5	5	8	9.75 ± 0.45	9.44 ± 0.42
13	50.5	3	4.5	10.20 ± 0.53	9.43 ± 0.40
14	50.5	3	4.5	11.46 ± 0.72	10.26 ± 0.68
15	50.5	3	4.5	8.48 ± 0.11	6.85 ± 0.51

**Table 4 foods-11-01257-t004:** ANOVA analysis of the percolation step.

	Source	Sum of Squares	df	Mean Square	F-Value	*p*-Value
BA	Model	20,379.64	9	2264.40	242.09	<0.0001
	X_1_—Volume	14,607.23	1	14,607.23	1561.70	<0.0001
	X_2_—pH	141.43	1	141.43	15.12	0.0115
	X_3_—Flow	8.85	1	8.85	0.9465	0.3753
	X_1_X_2_	37.34	1	37.34	3.99	0.1022
	X_1_X_3_	22.90	1	22.90	2.45	0.1784
	X_2_X_3_	0.1561	1	0.1561	0.0167	0.9023
	X_1_²	5530.76	1	5530.76	591.31	<0.0001
	X_2_²	17.81	1	17.81	1.90	0.2261
	X_3_²	0.0810	1	0.0810	0.0087	0.9295
	Error	46.77	5	9.35		
	Lack of Fit	42.29	3	14.10	6.30	0.14
	Pure Error	4.48	2	2.24		
SA	Model	20,702.25	9	2300.25	272.02	<0.0001
	X_1_—Volume	14,677.20	1	14,677.20	1735.67	<0.0001
	X_2_—pH	120.29	1	120.29	14.22	0.0130
	X_3_—Flow	15.09	1	15.09	1.78	0.2391
	X_1_X_2_	72.70	1	72.70	8.60	0.0326
	X_1_X_3_	38.19	1	38.19	4.52	0.0869
	X_2_X_3_	0.0088	1	0.0088	0.0010	0.9755
	X_1_²	5734.84	1	5734.84	678.18	<0.0001
	X_2_²	5.34	1	5.34	0.6312	0.4629
	X_3_²	0.0000	1	0.0000	5.390	0.9982
	Error	42.28	5	8.46		
	Lack of Fit	35.96	3	11.99	3.80	0.21
	Pure Error	6.32	2	3.16		

**Table 5 foods-11-01257-t005:** Experimental plan and results of the BBD for the elution step.

Experiment	X_1_ (%)	Experimental Plan		Log (Re)	
		X_2_ (mL)	X_3_ (mL min^−1^)	BA	SA
1	70	5.5	4.5	0.57 ± 0.023	0.58 ± 0.033
2	50	5.5	8	0.49 ± 0.012	0.50 ± 0.027
3	90	1	4.5	2.02 ± 0.032	2.01 ± 0.034
4	70	5.5	4.5	0.52 ± 0.013	0.58 ± 0.016
5	70	10	8	0.01 ± 0.009	0.09 ± 0.004
6	50	10	4.5	0.04 ± 0.007	0.04 ± 0.001
7	50	1	4.5	1.41 ± 0.017	1.52 ± 0.023
8	90	5.5	8	0.54 ± 0.013	0.60 ± 0.011
9	90	5.5	1	0.61 ± 0.017	0.65 ± 0.013
10	70	1	8	1.95 ± 0.027	1.91 ± 0.024
11	50	5.5	1	0.47 ± 0.012	0.56 ± 0.017
12	90	10	4.5	−0.05 ± 0.008	0.02 ± 0.004
13	70	5.5	4.5	0.40 ± 0.011	0.50 ± 0.012
14	70	10	1	−0.10 ± 0.007	0.06 ± 0.013
15	70	1	1	1.97 ± 0.014	1.98 ± 0.028

**Table 6 foods-11-01257-t006:** ANOVA analysis of the SPE elution step.

	Source	Sum of Squares	df	Mean Square	F-Value	*p*-Value
**BA**	Model	7.89	9	0.8762	62.90	0.0001
	X_1_—% MeOH	0.0784	1	0.0784	5.63	0.0437
	X_2_—Volume	7.13	1	7.13	512.08	<0.0001
	X_3_—Flow Rate	0.0001	1	0.0001	0.0097	0.9254
	X_1_X_2_	0.0971	1	0.0971	6.97	0.0460
	X_1_X_3_	0.0024	1	0.0024	0.1744	0.6936
	X_2_X_3_	0.0053	1	0.0053	0.3840	0.5626
	X_1_²	0.0080	1	0.0080	0.5707	0.4840
	X_2_²	0.5376	1	0.5376	38.59	0.0016
	X_3_²	0.0222	1	0.0222	1.59	0.2626
	Error	0.0696	5	0.0139		
	Lack of Fit	0.0568	3	0.0179	2.27	0.3207
	Pure Error	0.0158	2	0.0079		
**SA**	Model	7.35	9	0.8164	102.17	<0.0001
	X_1_—% MeOH	0.0539	1	0.0539	6.75	0.0484
	X_2_—Volume	6.69	1	6.69	837.35	<0.0001
	X_3_—Flow Rate	0.0000	1	0.0000	0.0028	0.9595
	X_1_X_2_	0.0652	1	0.0652	8.16	0.0356
	X_1_X_3_	0.0000	1	0.0000	0.0059	0.9416
	X_2_X_3_	0.0128	1	0.0128	1.60	0.2612
	X_1_^2^	0.0024	1	0.0024	0.2945	0.6107
	X_2_^2^	0.5108	1	0.5108	63.93	0.0005
	X_3_^2^	0.0109	1	0.0109	1.36	0.2958
	Error	0.0399	5	0.0080		
	Lack of Fit	0.0357	3	0.0119	5.56	0.1562
	Pure Error	0.0043	2	0.0021		

**Table 7 foods-11-01257-t007:** Optimal conditions for the SPE of BA and SA.

	Factor	Optimal Value	R%
**Percolation Step**	Sample volume (mL)	1	~99
	pH	1	
	Flow rate (mL min^−1^)	4.5	
**Elution Step**	MeOH percentage (%)	90	~95
	Eluent volume (mL)	1	
	Flow rate (mL min^−1^)	4.5	

**Table 8 foods-11-01257-t008:** Calibration equation, correlation coefficient, and detection and quantification limits of benzoic and sorbic acids.

	Linear Range (µg mL^−1^)	Calibration Equation Y = ax + b	R^2^	LDD (µg mL^−1^)	LDQ (µg mL^−1^)
**BA**	1–5	y = 27.537x + 0.2850	0.9646	0.177	0.592
	5–15	y = 65.889x − 92.109	0.9889		
	25–45	y = 43.030x − 65.610	0.9890		
**SA**	1–5	y = 36.852x + 3.561	0.9938	0.502	0.873
	5–15	y = 115.01x − 185.44	0.9760		
	25–45	y = 80.260x − 251.2	0.9851		

**Table 9 foods-11-01257-t009:** Recovery percentages (Rs%) of AB and AS from drink foods.

Sample		BA	SA
**Fruit Juice**	Sample 1	92.26 ± 1.02	82.99 ± 1.13
	Sample 2	90.58 ± 0.95	81.78 ± 1.04
	Sample 3	97.42 ± 2.04	94.57 ± 2.41
**Soft Drink**	Sample 1	74.43 ± 0.56	81.18 ± 1.01
	Sample 2	99.51 ± 2.54	96.13 ± 2.72
	Sample 3	98.21 ± 2.23	96.87 ± 2.91

**Table 10 foods-11-01257-t010:** Recovery percentages and the relative standard deviations (RSD%) of the three SPE methods tested.

	AC		MWCNT		Silica-Based C18	
	R (%)	RSD (%)	R (%)	RSD (%)	R (%)	RSD (%)
**BA**	67.62	1.85	75.25	5.67	98.23	1.64
**SA**	54.38	2.57	69.81	6.85	95.13	1.78

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
