# Peer review of "Application of Response Surface Methodology to Optimize Solid-Phase Extraction of Benzoic Acid and Sorbic Acid from Food Drinks"

_foods, 2022, doi:10.3390/foods11091257_

Round 1
Reviewer 1 Report
The objective of this study was to develop a mathematical model, for the optimization of the extraction process of two preservatives, benzoic and sorbic acids (BA, SA), from food drinks.
Thematically the work is interesting for the researchers and professionals and the proposed manuscript is relevant to the scope of the journal.
I found it appropriate for the Foods journal, but only after some modifications and clarification from the Authors.
The overall organization and structure of the manuscript are appropriate.
The paper is well written and the topic is appropriate for the journal.
The aim of the paper is well described and the discussion was well approached, its results and discussion are correlated to the cited literature data.
The literature review is comprehensive and properly done.
The novelty of the work must be more clearly demonstrated.
The significance of the Work: Given the large number of analyzed data, this is an interesting study with a possible significant impact in this area.
Statistical interpretation of the analytical data must be more properly presented: the verification of the model should be performed.
The coefficients in the regression models should be presented with SD values?
Figures 5 and 10 is not representative? Could you show a closer look, for instance log yield axis within the range 0-1, instead -0.5-2.5.
Other Specific Comments: The work is properly presented in terms of the language. The work presented here is very interesting and well done, it is presented in a compact manner.
Reviewer 2 Report
The manuscript entitled “Application of Response Surface Methodology to optimize Solid Phase Extraction of benzoic acid and sorbic acid from food drinks” developed a simple and reliable SPE method for simultaneous extraction of sorbic acid and benzoic acid from food drinks. The authors have presented good research scope and method optimization, however, the novelty statement was not very good and comprehensive. My comments listed as follow:
- Among the whole work, the authors do hard works for experimental optimization, but the novelty statement was absent in this manuscript. The reviewer suggests that the novelty of this work should be mentioned in the last paragraph of introduction. Moreover, the novelty should also be included in Abstract and Conclusions.
- Introduction: the reported sample preparation methods for the preservatives should be discussed more deeply and comprehensively. Indeed, organic solvent consumption in SPE is not so low, it is still larger than SPME.
- The maximum residue limits (MRLs) of the preservatives in different countries need to be pointed out in the introduction.
- When performing comparative tests, it should be noted that other SPE methods should also be performed under their optimized conditions. The authors should supply these experimental condition as well as their sources.
- Except the optimized factors, other important factors affecting the SPE performance, such as the mass of sorbent and the pH of acidified water in the eluent also need to be optimized.
- Table 1: X1 Sample volume was investigated from 1 to 100 mL. What is the reason for the investigation in such a wide range?
- Line 27-29, the optimized SPE condition: the sample volume and the sample pH in SPE percolation step, the percentage of MeOH and the solvent elution volume in SPE elution step were all set in the minimum or maximum value of the optimization range. Whether the experimental results would be better, if these condition was chosen at a smaller or larger values?
- Which points are predicted in Figures 1 and 2? Which points are observed?
- Some formatting and spelling problems:
-Tables should be display on the same page, not across pages. The caption of Fig. 7 was also located in the next page. It is not convenient for readers.
-Figure 3, left, it is BA instead of B.
-Line 376, Line 73: the ending lacks punctuation.
-Line 26: this should be This.
-Line 23-24: So, the … , were optimized. This sentence should be rewritten.
-Keywords: RSM should be used as full name. Also, Line 88-89, MWCNT and AC should be used as full name.
-Line 101, Line 106: mL min-1 and μg mL-1. Here, -1 need to be superscript.
-Line 193: CA should be AC.
-Line 269: figure should be Figure. Line 358: table should be Table.
-Line 371-374: The line spacing was different than elsewhere.
-The caption of Fig.11, Fig. 12 and Table 8: AB and AS should be BA and SA.
Reviewer 3 Report
The manuscript describe the development and optimization of a solid phase procedure for the determination of benzoic and sordic acids in food drinks. This issue is interesting but the some points as described below must be improved.
Introduction - Describe the state of art of analytical method used for analyzing these compounds. Why is necessary a new method? What are the advantages?
Materials and Methods. Several points must be improved.
Line 84 - What is the pH buffer solution ?
Line 87 and 88 –MWCNT and AC - explain what the acronym means
Line 96. Add version of software.
Line 99 – Use percentage to describe the mobile phase composition instead of volume
Line 109 – What is the concentrations of the HCl solution and NaOH solutions?
The validation procedure must be included in this section. What the guideline used? Add the reference.
Results and Discussion
The author reported three linear range ( 1-5; 5-15; 25-45 µg mL) – the author must be considere only one linear range.
The validation must be studied at least two spiked levels. The author must be explain how it was stablished the matrix effect, detection limit and quantification limit.
The authors should include the chromatograms of standards solution in mobile phase of the analytes used. What is the retention time of each compound?
The was coelution of interferents? Identify in figures 11 e 12 each peak showed in the chromatograms.
Why the extraction procedure was compared with AC and MWCNT spe cartridges – add references about that.
Is there an official method to analyze these compounds? What the advantages this sample preparation in regards to other analytical methods?
How many replicates ( for each SPE methods)were carried on in comparative study? Were they
sufficient and appropriate to reach the conclusion?
The LLDD and LDQ values were enough to met legal requirements?
Reviewer 4 Report
In the manuscript titled " Application of Response Surface Methodology to optimize Solid Phase Extraction of benzoic acid and sorbic acid from food drinks", the authors investigate the potential of solid phase extraction for the extraction of benzoic acid and sorbic acid from food drinks. The paper is interesting, but very poorly written and not suitable for publication in its present form. It lacks any discussion.
Specific comments are listed below:
-Abstract: Please avoid using abbreviations in the abstract.
-Introduction: lines 64-66 Please provide more information about the methods listed. What are their advantages, disadvantages, or limitations?
The motivation for the research and the novelty of the research should also be highlighted in the introduction.
Line 103.Please avoid abbreviations in section headings.
Line 107. if units of measurement are written with exponents, please check that all exponents are spelled correctly
Lines 115 through 138 contain too much theoretical information and can therefore be removed from the Materials and Methods section.
Tables 3 and 4 can be merged to avoid repetition
Table 4. experimental results should be reported with standard deviations. There is no discussion of the results presented or comparison with the available literature
Figure 1 and 2. Why are some data shown with blue dots and others with orange dots. Both figures are of poor quality. It is impossible to read the data on the axes.
Figure 3-10 poor quality. It is impossible to read the data on the axes.
Table 7. font too large. The experimental results should be given with standard deviations.
Different sizes of fonts in the text.
Line 349. how were the optimal conditions estimated? Please describe with more details.
Table 7. eq.9 Please use dot as singular for multiplication.
Round 2
Reviewer 2 Report
This manuscript still needs to be revised. My comments listed as follow:
- From the revised introduction, the novelty of this work seems insufficient. The authors should point out which point is the innovation of this work, and discuss its advantages.
- Please response comment 5 in previous comments, but not ignore it.
Except the optimized factors, other important factors affecting the SPE performance, such as the mass of sorbent and the pH of acidified water in the eluent also need to be optimized.
- Line 77 “opti-1mized”, please correct. And there are many other spelling and editing errors, please correct them carefully.
Author Response
the coorections are included in the attached file

Reviewer 3 Report
The manuscript was improved, but some aspects must be revised.
Line 51. The 5 maximum permitted levels in soft drinks is defined by the FAO/WHO Expert Committee at 150 mg/L – What compound? BA or SA?
Line 71. Therefore, , SPE has become an interesting - Correct the use of the comma
Line 89 “ Silica-based sorbent with octadecyl functional group was acquired from Applied Separations Company, Allentown, PA, USA (3 90 mL; 500 mg, with a particle size of 40 μm and an average porosity of 60 A°) - Is this the cartridge used for solid-phase extraction?
Line 464. The validation procedure must be improved. Usually two concentrations (at least) must be available, with five replicates. In general one of them is next to the LDQ value and other, is next to the maximum permitted level, in this case, 150 mg L-1. I understand that represest a wide concentration range, but is very important to evaluable the method in the maximum permitted level,
The LDQ values were 0.592 µg mL-1 and 0.873 µg mL-1 for BA and SA respectively. However the linear range both of them started in 1 µg mL-1 . Explain.
The real samples must be analyzed ( not only for comparative studies) and results (concentration of BA and SA) must be included in this manuscript.
Reviewer 4 Report
The authors did not marked the changes in the revised manuscript. In my option manuscript should still be revised in details before considering for publication.
The motivation for the research and the novelty of the research are still not clear
Experimental results should be reported with standard deviations to confirm the reliability of the measuring method.
Equations are of poor quality
Sizes of fonts in the text are not according to template
There are numerous typo in the manuscript
Line 223. The graph covers the text
Author Response
all corrections required are cited in the attached file and the manuscript
